# Optimization of Biotinylated RNA or DNA Pull-Down Assays for Detection of Binding Proteins: Examples of IRP1, IRP2, HuR, AUF1, and Nrf2

**DOI:** 10.3390/ijms24043604

**Published:** 2023-02-10

**Authors:** Yoshiaki Tsuji

**Affiliations:** Toxicology Program, Department of Biological Sciences, North Carolina State University, Campus Box 7633, Raleigh, NC 27695, USA; ytsuji@ncsu.edu

**Keywords:** RNA pull-down, DNA pull-down, biotin, streptavidin, IRP2, HuR, AUF1, Nrf2, ferritin, IRP-IRE

## Abstract

Investigation of RNA- and DNA-binding proteins to a defined regulatory sequence, such as an AU-rich RNA and a DNA enhancer element, is important for understanding gene regulation through their interactions. For in vitro binding studies, an electrophoretic mobility shift assay (EMSA) was widely used in the past. In line with the trend toward using non-radioactive materials in various bioassays, end-labeled biotinylated RNA and DNA oligonucleotides can be more practical probes to study protein–RNA and protein–DNA interactions; thereby, the binding complexes can be pulled down with streptavidin-conjugated resins and identified by Western blotting. However, setting up RNA and DNA pull-down assays with biotinylated probes in optimum protein binding conditions remains challenging. Here, we demonstrate the step-by step optimization of pull-down for IRP (iron-responsive-element-binding protein) with a 5′-biotinylated stem-loop IRE (iron-responsive element) RNA, HuR, and AUF1 with an AU-rich RNA element and Nrf2 binding to an antioxidant-responsive element (ARE) enhancer in the human ferritin H gene. This study was designed to address key technical questions in RNA and DNA pull-down assays: (1) how much RNA and DNA probes we should use; (2) what binding buffer and cell lysis buffer we can use; (3) how to verify the specific interaction; (4) what streptavidin resin (agarose or magnetic beads) works; and (5) what Western blotting results we can expect from varying to optimum conditions. We anticipate that our optimized pull-down conditions can be applicable to other RNA- and DNA-binding proteins along with emerging non-coding small RNA-binding proteins for their in vitro characterization.

## 1. Introduction

Understanding the regulatory mechanisms of gene expression is vital for deciphering cellular responses to a plethora of external cues and physiological conditions. Gene expression is regulated at the transcriptional, post-transcriptional, and translational levels as a result of changes in interactions between cis-acting regulatory elements of DNA and RNA with specific trans-acting proteins [1,2,3]. These core mechanisms of gene expression are also subject to epigenetic regulation, such as DNA methylation and histone modifications as well as modulations by micro RNAs (miRNAs), long non-coding RNAs (lncRNAs), and circular RNAs (circRNAs) [4,5,6]. In the past, the initial in vitro characterization of interactions between defined cis-acting elements and their binding proteins was carried out using electrophoretic mobility shift assays (EMSAs), in which ^32^P-end-labeled RNA or DNA harboring the defined sequence of interest was incubated with cell lysates, giving rise to the retarded migration of the radiolabeled RNA or DNA probe in non-denatured polyacrylamide gel electrophoresis upon the binding of proteins to the probe [7]. Along with the development of the non-radioactive detection of molecular interactions in bioluminescence resonance energy transfer (BRET) [8], force spectroscopy (FS) [9], molecular recognition imaging (MRI) [10], fluorescence cross-correlation spectroscopy (FCCS) [11], and EMSA [12,13], end-labeled biotinylated RNA and DNA oligonucleotides have been favorably used as non-radioactive probes in pull-down assays. The protein–RNA and protein–DNA binding complexes can be pulled down with streptavidin-conjugated resins for the subsequent identification of binding proteins by Western blotting or proteomic analyses. Despite the protocols of biotinylated RNA pull-down assays published by a couple of groups [14,15,16], it has not been demonstrated how to set up optimum pull-down reactions and what results we can expect. To fill this gap, each experiment in this study was designed to address key technical questions in RNA and DNA pull-down assays: (1) what amount of RNA or DNA probes is sufficient and optimum in the binding reaction; (2) what binding buffer we can use; (3) what lysis buffer we can use to prepare whole-cell lysates (WCLs); (4) how to verify the specific interaction; (5) what streptavidin resin (agarose or magnetic beads) works; and (6) what Western blotting results we can expect from varying conditions we would use for the characterization of RNA- and DNA-binding proteins.

This study shows our pull-down set-up and results for the detection of RNA- and DNA-binding proteins in the human ferritin H gene. Ferritin is the major intracellular iron storage protein composed of 24 subunits of heavy (H) and light (L) chains [17]. Iron homeostasis is tightly maintained by the coordinate balance between iron transport and storage in response to the fluctuation of iron availability [18,19,20]. The human ferritin H gene has at least three cis-acting elements: a 5′-untranslated RNA element termed the IRE (iron-responsive element) [21,22], a 3′-untranslated AU-rich RNA element characterized in this study, and a far-upstream DNA enhancer termed the antioxidant-responsive element (ARE), which we previously identified and characterized [23,24,25].

The IRE has a stem-loop RNA structure in the 5′-untranslated region (UTR) of several key iron metabolism genes including ferritin H, L, and an iron exporter ferroportin (SLC40A1), while the highly conserved IRE was also found in the 3′-UTR of mRNAs encoding iron transport protein transferrin receptor-1 (TfR1) and divalent metal transporter 1 (DMT1, SLC11A2) [22,26]. All these IREs bind two RNA-binding proteins, termed IRP1 and IRP2 [27]. The IRP-IRE interaction is inversely regulated by intracellular iron levels; namely, iron depletion enhances and iron repletion impairs the binding of IRPs to IREs [22,27]. In iron-deficient conditions, IRPs bound to the 5′-UTR IRE cause translational block [28], resulting in the downregulation of ferritin and ferroportin proteins to downsize the capacity of iron storage into ferritin and iron export by ferroportin. Reciprocally, IRPs bound to the 3′-UTR IRE under iron deficiency increase the stability of corresponding mRNAs, resulting in increased iron transport via the upregulation of TfR1 and DMT1 mRNAs [29,30]. This coordinated regulatory mechanism termed the IRP-IRE system [26] also allows cells to manage iron excess conditions by facilitating the dissociation of IRPs from IREs. The IRP-free IRE increases the translation of 5′-IRE-regulated ferritin and ferroportin for more iron storage and export, while the inverse 3′-IRE-mediated regulation through the degradation of TfR1 and DMT1 mRNAs halts iron transport [18,31].

The second cis-acting sequence in the human ferritin H mRNA is an AU-rich element located in the 3′-UTR as reported previously [32]. We also reported that the expression of ferritin H is regulated through mRNA stability in response to calcium elevation [33]. There are several putative AU-rich elements in the 3′-UTR of ferritin H mRNA (NCBI NM_002032.3). We demonstrated in this study by RNA-pull-down assays that one of the ferritin H AU-rich elements binds HuR and AUF1, suggesting their roles in the stability regulation of the ferritin H mRNA.

The third cis-acting element is located 4.5 kb upstream from the transcription start site of the human ferritin H gene, where we identified an antioxidant-responsive element (ARE) [23,24]. Nrf2 is recruited to the ARE along with additional b-zip family members in response to oxidative stress inducers such as arsenic and tert-butyl hydroquinone (t-BHQ) [24,25,34,35].

This work is focused on the in vitro characterization of binding proteins to cis-acting elements by setting up an optimal pull-down condition using biotinylated RNA and DNA oligonucleotides with streptavidin-conjugated resins. Since non-AGO-family RNA-binding proteins are known to directly interact with specific microRNAs and regulate targeted genes [36,37], we also discuss the possibility of the application of our pull-down assay conditions for the identification of proteins interacting with emerging non-coding small RNAs and beyond [38].

## 2. Results

### 2.1. The IRP-IRE Pull-Down Set-Up and Comparison between Streptavidin-Agarose and Magnetic Beads

The binding of IRP1 and IRP2 to IRE has been assessed in vitro traditionally by EMSA using a ^32^P end-labeled IRE RNA probe [39] and also recently probed with a fluorescent-labeled IRE RNA [13]. The identification of binding proteins in EMSA usually needs further characterization through a specific antibody-mediated inhibition or retardation of the protein binding complex [34], while RNA pull-down assays allow the identification of targeted proteins in the binding complex by Western blotting [14,15,16] or more unbiased proteomic approaches [40,41]. To optimize and establish the radioisotope-free and user-friendly IRP-IRE binding assay, we attempted to set up an RNA pull-down assay using 5′- biotinylated 32nt human ferritin H IRE as a probe (wild-type and mutant IRE RNA oligonucleotides, Sigma-Aldrich, St. Louis, MO, USA, Figure 1A). First, as magnetic beads are much easier for washing and the subsequent recovery of precipitates, we compared streptavidin magnetic beads to high-capacity streptavidin agarose resin. An amount of 2 μg of the biotinylated IRE RNA (wt) was incubated with whole-cell lysates (WCL) from SW480 human colon carcinoma cells in magnetic beads (MBs) binding buffer (for MBs) or binding buffer A (for agarose) at room temperature for 1 h, followed by a further 1 h incubation with 2.5–20 μL suspensions of streptavidin magnetic beads or 20 μL of streptavidin agarose (Figure 1B). Both beads were pre-washed once with modified RIPA buffer (see washing buffer in table in Section 4) and resuspended in each binding buffer prior to the incubation. The pull-down samples were subjected to IRP2 Western blotting. More details on the pull-down procedures and reagents can be found in the Materials and Methods section, each figure legend, and table in Section 4. Unexpectedly, we experienced very poor pull-down of the IRP2-IRE complex with streptavidin-magnetic beads (2.5–20 μL) in the MB binding buffer recommended by NEB (20 mM Tris-HCl pH7.5, 0.5 M NaCl, 1 mM EDTA) while 20 μL streptavidin-agarose in the binding buffer A (20 mM Tris, pH7.4, 300 mM KCl, 1.5 mM MgCl_2_, 0.2 mM EDTA, 0.5 mM PMSF) pulled down the IRP2-IRE complex much more efficiently (Figure 1B).

We tested the possibility that the failure of IRP2-IRE pull-down with streptavidin magnetic beads might be due to the binding buffer difference. The magnetic beads failed to pull down the IRP2-IRE binding complex in the MB buffer again (Figure 1C). Streptavidin-agarose in the MB buffer also lost the pull-down capability compared to the same amount (20 μL) of streptavidin-agarose in binding buffer A (Figure 1C). If the MB buffer would be the general problem for the pull-down of the IRP-IRE binding complex, we anticipated that the streptavidin-magnetic beads in buffer A or buffer C we used for EMSA may improve the pull-down efficiency. However, none of these buffers improved pull-down with streptavidin-magnetic beads even when twice the amount of magnetic beads (40 μL) was used (Figure 1D, lanes a–f). In contrast, streptavidin-agarose (20 μL) in buffer A as well as buffer C precipitated the IRP2-IRE binding complex (Figure 1D, lanes g and h). As another streptavidin-magnetic beads in buffer A or C failed to pull down the IRP2-IRE complex (see Dynabeads M280 data in Figure 2D), we concluded that our experimental conditions were not optimum for the magnetic beads we tested in this IRE pull-down application (see Section 3). We therefore focused on high-capacity streptavidin-agarose (ThermoFisher Scientific, Waltham, MA, USA) for the further optimization of RNA and DNA pull-down assays.

An amount of 20 μL of high-capacity streptavidin-agarose (binding capacity: >10 μg biotinylated BSA/μL resin) was used for 500 μg of WCLs to pull down the IRP2-IRE complex in Figure 1B–D. However, to assess the appropriate amount of streptavidin-agarose for IRE pull-down assays, we tested varying amounts (5–30 μL) of high-capacity streptavidin-agarose to precipitate the IRP2-IRE complex. The IRP2 Western blot in Figure 1E showed that even a 5 μL suspension of high-capacity streptavidin-agarose was sufficient and equivalent to a 20 or 30 μL suspension to pull down the complex. As the packed precipitate of a 5 μL suspension of streptavidin agarose was too small to be easily handled during wash without loss of resins, we decided to use 20 μL of streptavidin-agarose for the rest of the pull-down assays.

The next question was whether this IRP2-IRE interaction is specific or not. To address this important issue, we used a mutated IRE RNA (Figure 1A) as a probe and a competitor as well. The biotinylated wtIRE reproducibly precipitated IRP2 whereas the mutant IRE failed to pull down IRP2 (Figure 1F). Furthermore, IPR2 bound to biotinylated wtIRE was competed out with non-biotinylated wtIRE RNA in a 4-fold excess condition, while the non-biotinylated mutant IRE failed in the same competition condition (Figure 1F). These results suggest that our pull-down set-up detects the sequence-specific interaction between the biotinylated IRE probe and IRP2.

The next important questions were (1) what amount of biotinylated IRE RNA is sufficient and optimum to pull down the IRP-IRE complex and (2) whether this pull-down assay is semi-quantitative. To address the first question, we tested 1, 2, 4, and 6 μg (100–600 pmol) of biotinylated wtIRE probe for 500 μg of SW480 WCL in the 200 μL binding buffer A, followed by pull-down with 20 μL of high-capacity streptavidin agarose. As shown in Figure 2A, 1 μg of the biotinylated IRE probe was sufficient for the binding of IRP2 as well as IRP1, and the increase in the probe up to 6 μg had no additional pull-down effects (Figure 2A). We also observed the marginal effect by switching binding buffer A to buffer C with 4 μg of the IRE probe (Figure 2A). Similar results were obtained when 500 μg of K562 WCL was used (Figure 2B). Based on the results in Figure 2A,B, we used 2 μg (200 pmol) of a biotinylated RNA probe for the rest of the experiments.

To address the second question of whether the pull-down assay is semi-quantitative or not, we incubated 100, 250, and 500 μg (each duplicate) of WCLs from K562 and SW480 cells with 2 μg of a biotinylated IRE probe for 1 h followed by another 1 h incubation with 20 μL of high-capacity streptavidin agarose to pull down the IRE binding complex. As shown in Figure 2C, we observed that increasing amounts of input WCLs gave rise to a more intensified IRP2 band. IRP1 also showed a similar trend but the IRP1 bands from 250–100 μg of K562 WCLs were too weak to be visualized (Figure 2C). Taken together, we concluded that our RNA pull-down assay using 2 μg of the biotinylated IRE probe and 20 μL of high-capacity streptavidin agarose is semi-quantitative enough to measure the binding of IRP2 and IRP1 to IRE.

To corroborate our conclusion in the IRE-IRP pull-down set-up, we prepared WCLs from K562 cells treated with 100 μM FAC (ferric ammonium citrate) or 25 μM iron chelator DFO (deferoxiamine mesylate) for 24 h and tested whether we could detect changes in the binding of IRP2 and IRP1 to the IRE under high- and low-iron conditions. We also tested another streptavidin-agarose (Invitrogen 15492-050, Waltham, MA, USA, equivalent to Life Technologies SA100-04 that binds 3–8 μg of biotinylated IgG per μL suspension) and another type of magnetic bead, Dynabeads M-280 (Invitrogen). As shown in Figure 2D, IRP2 binding to IRE was decreased following FAC treatment while it increased after iron chelator DFO treatment. The increased IRP1 binding to IRE was also observed in DFO-treated cells whereas the effect of FAC was marginal. Like NEB magnetic beads (Figure 1B–D), we failed to pull down the IRP-IRE complex with 30 μL of Dynabeads M-280 (Figure 2D).

### 2.2. Detection of Binding Proteins to the 3′-UTR AU-Rich Element in the Human Ferritin H mRNA

Little is known about binding proteins to the 3′-UTR of ferritin mRNA, partly due to the significant roles of the translational regulation via the IRP-IRE interaction in ferritin expression [42,43,44] along with the transcriptional regulation through the antioxidant-responsive element (ARE) [23,34,45]. It was shown by EMSA that U-rich sequences in the 3′-UTR of the human ferritin H mRNA bind unknown protein(s) that seemingly play a role in the post-transcriptional regulation of ferritin H in PMA-treated human monocytic THP1 cells [32]. We also reported that the expression of ferritin H is regulated via mRNA stability in response to calcium elevation [33]. There are at least four putative AU-rich elements in the human ferritin H mRNA (NM_002032), which harbor the core AU-rich sequence U(U/A)(U/A)UUU(U/A)(U/A)U [46]. We used one of them, located at nt1026 to nt1053 in NM_002032, as a biotinylated RNA probe (Figure 3A) for pull-down assays. Since AU-rich RNA sequences are known to bind several binding proteins including HuR and AUF1 [47], we applied our IRP-IRE pull-down conditions for pull-down assays of these RNA-binding proteins. Indeed, the AU-rich element in the ferritin H mRNA bound HuR and AUF1 in three human cell lines (Figure 3B). This interaction seemed to be specific to the AU-rich sequence because the pull-down of HuR was failed when the mutant probe with the impaired AU-rich sequence (6Us to 6Cs, Figure 3A) was used as a probe (Figure 3C). We concluded that our RNA pull-down conditions can be used not only for IRPs but also AU-rich element-binding proteins, and that the expression of ferritin H may be subject to mRNA stability regulation through AU-rich elements and their binding proteins including HuR and AUF1.

### 2.3. DNA Pull-Down for Detection of Nrf2 Bound to the Ferritin H Antioxidant-Responsive Element

In addition to the post-transcriptional regulation through these cis-acting RNA elements, both human and mouse ferritin H genes are transcriptionally regulated via the antioxidant-responsive element (ARE) enhancer [23,45], to which Nrf2 and other b-zip family members bind under chemical and oxidative stress conditions [24,25,34]. Our previous in vitro characterization of the ferritin H ARE-binding proteins was carried out by EMSA [34]. In this study, we applied the RNA pull-down procedures to DNA pull-down assays for the detection of Nrf2 bound to the ferritin H ARE. As shown in Figure 4A, the human ferritin H ARE is composed of a 22 bp AP1-like element (ARE1) and 23 bp AP1/NFE2 element (ARE2) separated by a 20 bp spacer [23,34]. Nrf2 was detected by Western blotting in 20 μg of the cytoplasmic and nuclear fractions of K562 cells treated with 25 μM sodium arsenite (NaAsO_2_) or 25 μM t-BHQ (tert-butyl hydroquinone) for 14 h (Figure 4B, input). To assess the binding of the activated Nrf2 to the AREs, 4 μg of annealed 5′-biotinylated sense and antisense strand oligonucleotides of ARE1 and ARE2 (2 μg each of the strand) were incubated with 100 μg of the K562 cytoplasmic and nuclear fractions. In this binding reaction, we simultaneously added all together with 30 μL of streptavidin-agarose (Invitrogen) in 200 μL of PBS (phosphate-buffered saline) as a binding buffer and incubated it with constant rotation at room temperature for 2 h. After spinning down the streptavidin-agarose and wash with PBS, ARE-binding proteins were eluted into 10 μL of 2xSDS-PAGE sample buffer by heating at 90 °C for 5 min and subjected to Western blotting for the detection of Nrf2. Consistently, this pull-down assay showed that both arsenic and t-BHQ treatments increased the binding of the nuclear Nrf2 to the ARE1 and ARE2 (Figure 4B). Not only PBS as a binding buffer but also buffer C, used in our EMSA [34] and the IRE-IRP pull-down in Figure 2A,B, worked well in the ARE1, ARE2, and complete ARE (65 bp) pull-down assays (Figure 4C).

## 3. Discussion

We demonstrated here how to set up biotinylated IRE and AU-rich RNA pull-down assays for IRP2, IRP1, HuR, and AUF1 in addition to biotinylated ARE DNA pull-down for Nrf2. In the IRE-IRP pull-down assay, we used 2 μg (200 pmol) of a 5′-biotinylated IRE RNA probe for 500 μg of SW480 or K562 WCL in a total 200 μL of binding buffer A or C, along with incubation with constant rotation for 1 h at room temperature, followed by another 1 h incubation with 20 μL of pre-washed high-capacity streptavidin agarose (binding capacity: >10 μg biotinylated BSA/μL resin, ThermoFisher Scientific). Alternatively, 30 μL of streptavidin agarose (binding capacity: 3–8 μg of biotinylated IgG per μL suspension, Invitrogen) worked well to pull down the biotinylated IRE-IRP binding complex. We noticed no clear difference in pull-down efficiency between binding buffer A and buffer C (Figure 1D and Figure 2A and the buffer compositions in table in Section 4). PBS was recommended for a binding buffer of high-capacity streptavidin agarose in batch analyses (ThermoFisher Scientific, also suggested to add 0.1 % SDS, 1% NP40, or 0.5% sodium deoxycholate to reduce non-specific binding). Although we did not try to use PBS for our RNA pull-down assays, note that PBS worked nicely in biotinylated DNA pull-down assays for the detection of Nrf2 binding to ARE (Figure 4). The successive incubation for 1 h each of a biotinylated RNA or DNA probe with WCL followed by the addition of streptavidin agarose may not be necessary because mixing the biotinylated probe, WCL, and streptavidin agarose all together for 1–2 h successfully pulled down the binding complexes of IRP-IRE (Figure 2D), HuR and AUF1 with an AU-rich RNA (Figure 3B,C), and Nrf2-ARE (Figure 4B,C). Furthermore, RNase and protease inhibitors, such as 10 mM of ribonucleotide vanadyl complex (NEB S1402S) and 1× protease inhibitor cocktail set I (Millipore Calbiochem 539131), can be added to the binding buffer; however, we did not observe the merit during 1–2 h incubation at room temperature for the pull-down of IRPs, HuR, AUF1, and Nrf2 as shown in this study. If the degradation of proteins would be an issue, we may be able to shorten the incubation time to 15–30 min as we did in EMSA [23,48]. We also noticed that there is no clear difference in the pull-down efficiency of WCLs prepared in IP lysis buffer or RIPA buffer (see figure legends for Figure 1). However, we recommend that the volume of WCL should not exceed ~20% (40 μL) of the final binding reaction volume (usually 200 μL in binding buffer A or C). If necessary due to the lower protein concentration of WCLs, we recommend just to increase the total binding reaction volume from 200 μL to 500 μL without increasing other components as the biotinylated RNA probe and streptavidin agarose are still sufficient (Figure 1E and Figure 2A,B).

To confirm the specificity of IRE-IRP interactions, we tested a 5′-biotinylated mutant IRE probe for binding to IRPs, in addition to a pull-down competition assay by adding 4-fold in excess of non-biotinylated wt and mutant IRE competitors (Figure 1F). Like EMSA, the inclusion of non-biotinylated wt and mutant competitors is important for the verification of their specific interactions between a probe and binding proteins in the pull-down assays.

We intended that the conditions optimized in this work can be used for RNA and DNA pull-down assays and detection by Western blotting to test (1) whether the binding protein predicted from the defined nucleotide sequences is present or not in the binding complex and (2) whether the protein binding is increased or decreased according to gene expression changes in the same experimental conditions. Protein purities in the pull-down samples are therefore not critical issues for such targeted pull-down assays detected by Western blotting with a specific antibody. However, we also tested the effect of pre-clearing WCLs with a biotinylated mutant IRE and streptavidin-agarose on the protein precipitates subsequently pulled down with a biotinylated wild-type IRE RNA probe. The results showed that the pull-down with a biotinylated IRE RNA probe per se (no pre-clear) can remove the majority of non-interacting proteins (compare lane 1 to 1/25 of input WCL in the Appendix A). Pre-clearing WCLs with a biotinylated mutant IRE or/and streptavidin agarose did not improve the efficiency of IRP2 pull-down and the purity of the pull-down protein samples (Appendix A).

We are not sure about the reason for the inability of the magnetic beads to pull down the IRP-IRE binding complex in our assays; however, some key procedures including the biotinylation of an RNA probe and/or binding conditions of the probe with magnetic beads and WCLs may need to be further optimized or modified. For instance, the Magnetic RNA-Protein Pull-Down Kit (ThermoFisher Pierce) uses a 3′-desthiobiotinylated RNA probe (RNA incubated with biotinylated cytidine bisphosphate and T4 RNA ligase) and incubates first with magnetic beads in RNA capture buffer (20 mM Tris pH7.5, 1 M NaCl, and 1 mM EDTA), followed by incubation with WCL in protein–RNA binding buffer (20 mM Tris pH7.5, 50 mM NaCl, 2 mM MgCl_2_, 0.1% Tween 20). As desthiobiotin binds streptavidin reversibly, HuR bound to an AU-rich RNA was gently eluted with 4 mM biotin-containing elution buffer [49]. Furthermore, it was reported that there are significant variations in the binding capacity of streptavidin magnetic beads not only from different vendors but also intralot numbers from the same vendor [50]. In addition, binding capacities of various streptavidin beads including those we used in this study were assayed for different proteins (not biotinylated RNA or DNA) [50]. This may make it more difficult to find the optimum of streptavidin beads when they are not working. Of note, using much more magnetic beads (100 μL of 10 mg/mL Dynabeads M-280) might be necessary for pull-down assays as recommended [51]; however, 100 μL of magnetic beads per sample is not practical for many laboratories due to the cost of magnetic beads per sample.

A growing number of studies on interactions between non-coding RNAs and macromolecules such as RNA, DNA, and proteins have been reported, and representative interactions with lncRNAs (long non-coding RNAs) are summarized by Kazimierczyk et.al [4]. For the characterization of proteins that specifically interact with lncRNA, mass spectrometry was employed after the pull-down of a biotinylated lncRNA and protein binding complex with streptavidin magnetic beads. For instance, using the Magnetic RNA-Protein Pull-Down Kit, the lncRNAs TPA [41], XIST [52], and RP3-326I-13.1 (PINCR) [53] were in vitro transcribed and desthiobiotin-labeled for pull-down with streptavidin magnetic beads and subjected to the identification of several interacting proteins such as HSP90B by mass spectrometry [41,53]. Similarly, ADAR1 (adenosine deaminase RNA specific 1) was identified as a binding protein to the immunosuppressive lncRNA LINC00624 that was in vitro transcribed in the presence of biotin-16-UTP and pulled down with Dynabeads C1 and mass spectrometry [40]. The LINC00624 binding to ADAR1 stabilized the ADAR1 protein that ultimately promotes tumor progression [40]. Proteins interacting with circular RNAs (circRNAs) produced during RNA splicing were also characterized by pull-down with a biotinylated probe and streptavidin magnetic beads [54].

Accumulating evidence has indicated that non-AGO-family RNA-binding proteins directly interact with specific microRNAs and thereby play important regulatory roles in gene expression and cell physiology [36] including cancer [37]. For instance, HuR was shown to block the miR-21-mediated translational repression of the tumor suppressor programmed cell death 4 (PDCD4) through two mechanisms: binding competition with miR-21 on the 3′-UTR of PDCD4 mRNA and direct interaction with miR-21 thereby working as a miR-21 sponge [55]. HuR was also shown to interact with miR-122 and facilitates the extracellular vesicle-mediated export of miR-122 in human liver cells under starvation [56]. The AUF1 p37 isoform was shown to bind miRNA let-7b and promote loading let-7b onto AGO2 for the enhanced repression of target mRNA [57]. In these studies, HuR, AUF1, and miRNA interactions were assessed or confirmed by EMSA [55,56,57]. We think that our biotinylated AU-rich RNA pull-down conditions for HuR and AUF1 can be applicable for biotinylated miRNA pull-down assays to assess miRNA interactions with HuR, AUF1, and other RNA-binding proteins.

In addition to biotinylation, various methods for RNA labeling to investigate RNA–protein interactions were summarized by Gemmill et al. [58] and further characterizations of RNA–protein interactions including RIP (RNA immunoprecipitation) and CLIP (cross-linking and immunoprecipitation) were reviewed by Barra and Leucci [59].

Conclusions and Future Perspectives: We determined the optimal condition for an IRP-IRE pull-down assay, in which 2 μg (200 pmol) of a 5′-biotinylated IRE RNA probe was used for 500 μg of WCL (either prepared in IP lysis buffer or RIPA) in a total of 200 μL of binding buffer A or binding buffer C. The incubation with constant rotation for 1 h at room temperature was followed by another 1 h incubation with 20 μL of pre-washed high-capacity streptavidin agarose or 30 μL of streptavidin agarose. Mixing all together and incubation for 1–2 h also worked well. To confirm the specificity of IRE-IRP interaction, the inclusion of 2 μg of a 5′-biotinylated mutant IRE probe compared to wt IRE probe in pull-down assays is important along with a competition assay by adding >4-fold in excess of non-biotinylated wt and mutant IRE competitors. This pull-down assay condition was applied for the detection of HuR and AUF1 bound to an AU-rich RNA element as well as Nrf2 bound to an ARE double-strand DNA enhancer element. The pull-down assays optimized in this work will save a fair amount of time for researchers who will need to establish and perform RNA and DNA pull-down assays for their initial characterization of binding proteins anticipated from their RNA and DNA sequences. We expect that this pull-down condition will also help or facilitate the characterization of macromolecules interacting with small non-coding RNAs that regulate the expression of target genes.

## 4. Materials and Methods

### 4.1. Cell Culture, Chemicals, and Buffer

All human cell lines (SW480, K562, HepG2, HaCaT, HEK293, Jurkat, and HL60) were cultured in a humidified 37 °C CO_2_ incubator (5% CO_2_, MCO-17AIC, Sanyo, Osaka, Japan). The culture media and chemicals used in this work are listed in Table 1 along with suppliers, catalog numbers, and lot numbers if applicable. Ferric ammonium citrate (FAC) and deferoxamine mesylate (DFO) were dissolved in purified water (Millipore Z00QSV001, Milli-Q^®^ system, MilliporeSigma, Burlington, MA, USA) at 100 mM and 25 mM, respectively. Sodium arsenite (NaAsO_2_) was dissolved in Milli-Q^®^ water at 10 mM. t-BHQ was freshly prepared, first dissolved in DMSO at 1 M and further diluted to 10 mM with Milli-Q^®^ water.

### 4.2. RNA and DNA Oligonucleotides

All biotinylated and non-biotinylated oligonucleotides were synthesized by Sigma-Aldrich (St. Louis, MO, USA). Desalt grade RNA oligonucleotides at 0.2 μmol scale yielded approximately 500–700 μg. We experienced that the synthesis scale increased to 1.0 μmol did not increase the yield of IRE RNA oligonucleotides. Desalt-grade DNA oligonucleotides at a 0.05 μmol scale yielded 200–500 μg. All RNA and DNA oligonucleotides were dissolved at 1 μg/μL in TE (10 mM Tris, pH7.4 and 1 mM EDTA). 32nt IRE RNA oligos at 1 μg/μL is approximately 100 μM. DNA oligonucleotides were annealed by mixing 100 μL each of 1 μg/μL sense and antisense 5′-biotinylated ARE oligonucleotides together with 20 μL of NEBuffer 3 (50 mM Tris-HCl, pH 7.9, 10 mM MgCl_2_, 100 mM NaCl, 1 mM DTT, New England Biolabs, Ipswich, MA, USA) in a microcentrifuge tube, followed by incubation in 300–400 mL of microwaved 90 °C water in a beaker and leaving it at room temperature until gradually cooling down to room temperature. The sequences of human ferritin H RNA and DNA oligonucleotides used in this study (biotinylation at the 5′-end) are as follows:

IRE (wt) RNA: 5′-GGUUUCCUGCUUCAACAGUGCUUGGACGGAAC-3′

IRE (mt) RNA: 5′-GGUUUCCUGCUUCAAGCUCCGUUGGACGGAAC-3′

AU-rich (wt) RNA: 5′-AAACGAGUAUUUGUAUUUAUUAAACUCA-3′

AU-rich (mt) RNA: 5′-AAACGAGUACCCGUACCCAUUAAACUCA-3′

ARE1 sense DNA: 5′-TCCTCCATGACAAAGCACTTTT-3′

ARE1 antisense DNA: 5′-AAAAGTGCTTTGTCATGGAGGA-3′

ARE2 sense DNA: 5′-GGAGTGCTGAGTCACGGTGGGA-3′

ARE2 antisense DNA: 5′-TCCCACCGTGACTCAGCACTCC-3′

Complete sense ARE: 5′-TCCTCCATGACAAAGCACTTTTGAGCCCAAGCCCAGCCTAGCGGAGTGCTGAGTCACGGTGGGA-3′

Complete antisense ARE 5′-TCCCACCGTGACTCAGCACTCCGCTAGGCTGGGCTTGGGCTCAAAAGTGCTTTGTCATGGAGGA-3′.

### 4.3. Streptavidin-Agarose and Streptavidin-Magnetic Beads

We used two streptavidin-agarose resins: high-capacity streptavidin-agarose (20359, ThermoFisher Scientific, Waltham, MA, USA) and streptavidin agarose (15942-050, Invitrogen, Waltham, MA, USA), which is the same product as currently available SA100-04. The resins were mixed well before being taken (20 μL × sample numbers) into a microcentrifuge tube, washed once with 1–1.5 mL of washing buffer (25 mM Tris, pH7.4, 15 mM NaCl, 1% NP40, and 0.5% sodium deoxycholate), and resuspended in (20 μL × sample numbers) of binding buffer A (20 mM Tris pH7.4, 300 mM KCl, 0.2 mM EDTA. 1.5 mM MgCl_2_, and 0.5 mM PMSF). Streptavidin magnetic beads (S1420S New England Biolabs (ΝΕΒ, Ipswich, MA, USA) and Dynabeads M-280 Invitrogen, Waltham, MA, USA) were also tested for IRE-IRP pull-down assays. These beads were magnetized with a DynaMag-2 magnetic stand during washing with NEB magnetic beads buffer (20 mM Tris-HCl pH7.5, 0.5 M NaCl, and 1 mM EDTA) or Dynabeads buffer (5 mM Tris-HCl pH7.5, 1 M NaCl, and 0.5 mM EDTA).

### 4.4. RNA and DNA Pull-Down Procedure

For the preparation of whole-cell lysates (WCLs) from SW480 and other adherent cells, subconfluent to confluent cells were washed with phosphate-buffered saline (PBS: 137 mM NaCl, 27 mM KCl, 15 mM KH_2_PO_4_, 81 mM Na_2_HPO_4_) and lysed in IP lysis buffer (25 mM Tris, pH7.4, 150 mM NaCl, 1 mM EDTA, 1% NP40, and 5% glycerol) or RIPA buffer (25 mM Tris, pH7.4, 150 mM NaCl, 1% NP40, 0.5% sodium deoxycholate, and 0.1% SDS). There was no difference in our results of RNA-pull down assays when WCL was prepared in IP lysis buffer or RIPA buffer. K562 and other suspension cells (0.5–1 × 10^6^ cells/mL) were centrifuged at 1000 rpm, and cell pellets were washed with PBS and lysed in IP lysis buffer. Protein concentrations in WCLs were 5–25 μg/μL, measured with a protein assay dye reagent (#5000006, BIO-RAD, Hercules, CA, USA). Except for the optimization of input probes and the assessment of semi-quantitative pull-down assays in Figure 1 and Figure 2, 500 μg of WCLs were incubated (rotated) at room temperature for 1 h with 2 μg of a biotinylated RNA probe in 200 μL of binding buffer A or binding buffer C (20 mM Hepes, pH 7.4, 100 mM KCl, 0.5 mM EDTA, 1.5 mM MgCl_2_, 20% glycerol, 1 mM DTT). There was no difference in the IRP pull-down efficiency incubated in binding buffer A or C (Figure 2A,B). In the IRE competition assay (Figure 1F), 2 μg (probe:competitor = 1:1) or 8 μg (probe:competitor = 1:4) of non-biotinylated wt or mt IRE oligonucleotide was added to the binding reaction. After the 1 h incubation for biotinylated RNA probe and protein interaction, 20 μL of high-capacity streptavidin-agarose (ThermoFisher Scientific, Waltham, MA, USA) or streptavidin agarose (Invitrogen, Waltham, MA, USA) was added to the binding reaction and further incubated for 1 h at room temperature. The binding reaction was terminated by centrifugation at 5000 rpm for 0.5 min and washing the resins with 1 mL of washing buffer twice. After the complete removal of the washing buffer, 12 μL of a 2xSDS-PAGE sample buffer (63 mM Tris, pH 6.8, 25% Glycerol, 2% SDS, 0.01% bromophenol blue, and 5% β-mercaptoethanol) was added to the resins, vortexed briefly, and heated at 95 °C for 10 min in a heating block dry bath (11-718-2, Fisher Scientific, Hampton, NH, USA) containing water.

For the ARE DNA pull-down assays, 50–75 μg of cytoplasmic and nuclear fractions was incubated (rotated) at room temperature for 1–2 h with 4 μg of annealed 5′-biotinylated ARE DNA in 200 μL of binding buffer C. The ARE DNA pull-down assay for the detection of Nrf2 was also performed in PBS as binding buffer (Figure 4B). The rest of the procedure is the same as the RNA pull-down assays using streptavidin agarose resins.

In IRE-IRP RNA pull-down assays for testing streptavidin magnetic beads (NEB, Ipswich, MA, USA) in Figure 1, binding buffer A, C, or NEB magnetic beads binding buffer was used as indicated. Buffer C was used for the binding buffer during the incubation with Dynabeads M-280 in Figure 2D. After the binding step, NEB magnetic beads and Dynabeads M-280 were magnetized and washed twice with NEB magnetic beads buffer and Dynabeads buffer, respectively. After the complete removal of the washing buffer, 12–15 μL of a 2xSDS-PAGE sample buffer was added to the beads, briefly vortexed, and heated at 95 °C for 10 min for the elution of the protein binding complex.

### 4.5. Western Blotting

After briefly spinning down heated samples in the microcentrifuge tubes, binding complexes eluted in the 2xSDS-PAGE sample buffer were loaded on a 10% acrylamide SDS-PAGE (10% acrylamide, 0.3% bisacrylamide, 375 mM Tris, pH 8.8, 0.1% SDS) mini-gel (4 × 2.9 inches, 0.75 mm thickness), along with protein size markers (Precision Plus Protein Standards, BIO-RAD, Hercules, CA, USA). The stacking gel was 5% acrylamide, 0.1% bisacrylamide, 125 mM Tris, pH 6.8, and 0.1% SDS. The electrophoresis was run at 15 mA per mini-gel for approximately 1.5 h in the SDS-PAGE running buffer (25 mM Tris base, 192 mM Glycine, and 0.1% SDS, no pH adjustment). Proteins separated on the SDS-PAGE gel were transferred to the PVDF membrane (Immobilon-P, IPVH00010, Millipore, Burlington, MA, USA) at 300 mA for 1.5 h in a mini trans-blot cell (022711PM, BIO-RAD, Hercules, CA, USA) filled with a Western blotting transfer buffer (25 mM Tris pH 8.3, 192 mM Glycine, and 20% methanol) at 4 °C. The PVDF membrane was soaked at room temperature for 15–30 min (blocking) in either 1% BSA (OmniPur, 2930, EMD Chemicals, Burlington, MA, USA) or 5% skim milk (232100, Difco^TM^, Beckton Dickinson, Franklin Lakes, NJ, USA) dissolved in 0.1% Tween-20-containing Tris-buffered saline (TBS: 20 mM Tris, pH 7.6 and 137 mM NaCl). Incubation with a primary antibody was conducted at 4 °C overnight on a rocker platform (Speci-Mix, Barnstead Thermolyne, Dubuque, IA, USA), followed by washing with 0.1% Tween20/TBS (15 min, 3 times) and incubation with a secondary antibody at room temperature for 1.5 h. All primary and secondary antibodies were diluted with the same blocking solution (either 1% BSA or 5% skim milk in Tween20/TBS). The PVDF membrane was washed three times, incubated with ECL reagents (Clarity, BIO-RAD, Hercules, CA, USA), and immediately exposed to X-ray films (ProSignal Blotting Film 30-810 L, Genesee Scientific, San Diego, CA, USA). For incubation of the same membrane with another primary antibody, the membrane was soaked in the stripping solution (1.5% glycine, 0.1% SDS, 1% Tween 20, pH 2.2 adjusted with HCl) for 1 h on a rocker and the incubation was repeated with primary and secondary antibodies.

### 4.6. Antibodies

The primary antibodies and conditions used in this study are as follows:

Anti-IRP2 mouse monoclonal antibody (SC-33682, lot K0409, Santa Cruz Biotechnology, Dallas, TX, USA), 500~1000-fold dilution with 1% BSA in Tween/TBS, 4 °C overnight.

Anti-IRP2 rabbit monoclonal antibody (D6E6W, lot 1, Cell Signaling Technology, Danvers, MA, USA), 1000-fold dilution with 1% BSA in Tween/TBS, 4 °C overnight.

Anti-IRP1 rabbit monoclonal antibody (ab126595, lot YI071316CS, abcam, Waltham, MA, USA), 1000-fold dilution with 5% skim milk in Tween/TBS, 4 °C overnight.

Anti-AUF1 rabbit monoclonal antibody (D604F, lot 1, Cell Signaling Technology, Danvers, MA, USA), 1000-fold dilution with 5% skim milk in Tween/TBS, 4 °C overnight.

Anti-HuR rabbit monoclonal antibody (D9W7E, lot 1, Cell Signaling Technology, Danvers, MA, USA), 1000-fold dilution with 5% skim milk in Tween/TBS, 4 °C overnight.

Anti-Nrf2 rabbit polyclonal antibody (SC-13032x, lot I1311, Santa Cruz Biotechnology, Dallas, TX, USA), 1000-fold dilution with 5% skim milk in Tween/TBS, 4 °C overnight. As SC-13032 was discontinued, anti-Nrf2 rabbit monoclonal antibody (D1Z9C, lot 8, Cell Signaling technology, Danvers, MA, USA) may be able to be used at 1000-fold dilution with 5% skim milk in Tween/TBS, 4 °C overnight. However, transiently transfected Nrf2 was detected much stronger with SC-13032 than D1Z9C (Appendix A).

The HRP-conjugated secondary antibodies used in this study were either anti-rabbit IgG (AP132P, 500 μg/mL, MilliporeSigma, Burlington, MA, USA) or anti-mouse IgG (7076S, Cell Signaling Technology, Danvers, MA, USA) at 5000-fold dilution with the same blocking solution as used for the primary antibody.

### 4.7. Isolation of Cytoplasmic and Nuclear Fractions

For the DNA pull-down assay to detect Nrf2 binding to the ARE, we used cytoplasmic and nuclear fractions isolated from K562 cells. After treatment with NaAsO_2_ or t-BHQ, cells were washed with PBS, resuspended in 200–500 μL hypotonic buffer (20 mM Tris, pH 7.4, 10 mM NaCl, 3 mM MgCl_2_), and incubated for 15 min on ice. The 1/20 volume (e.g., 25 μL for 500 μL cell suspension) of 10% NP40 was added, vortexed for 10 s at a high setting, and the suspension was centrifuged at 12,000 rpm for 30 s (MX-160 microcentrifuge, TOMY TECH/AMUZA, San Diego, CA, USA). The supernatant was transferred to a microcentrifuge tube (cytoplasmic fraction). The nuclear pellet was suspended in 50–200 μL of the nuclear extraction buffer (100 mM Tris, pH 7.4, 2 mM Na_3_VO_4_, 100 mM NaCl, 1% Triton X-100, 1 mM EDTA, 10% glycerol, 1 mM EGTA, 0.1% SDS, 1 mM NaF, 0.5% deoxycholate, 20 mM Na_4_P_2_O_7_, and 1× proteinase inhibitor cocktail), vortexed for 10 s at high setting, and incubated on ice for more than 30 min. The nuclear pellet was vortexed for 30 s at a high setting (making bubbles is no problem) and centrifuged at 12,000 rpm for 10 min at 4 °C. The supernatant (nuclear fraction) was transferred to a new microcentrifuge tube. Protein concentrations in cytoplasmic and nuclear fractions were measured with the BIO-RAD protein assay dye reagent, usually yielding 5–15 μg/μL.

### 4.8. Plasmid DNA and Transfection

We prepared WCLs from HEK293 cells transiently transfected with pCMVSport6-IRP1 (MHS6278-202757152, Open Biosystems/ThermoFisher Scientific, Waltham, MA, USA) and pRK5-IRP2 plasmids for Western blotting positive control. pRK5-IRP2 was constructed from pCR4-TOPO-IRP2 (MHS4426-99239389, Open Biosystems/ThermoFisher Scientific, Waltham, MA, USA) by two steps: the ligation of IRP2 cDNA (SpeI/NotI) to pCRII SpeI/NotI sites (pCRII IRP2) followed by the isolation of IRP2 cDNA (BamHI/XbaI) and cloning to BamHI and XbaI sites of the pRK5 vector. pRK5Nrf2 was constructed by the ligation of a BamHI- and XhoI-digested Nrf2 DNA from pOBT7Nrf2 (ID#4548874, Invitrogen, Waltham, MA, USA) into BamHI and SalI sites of the pRK5 vector. These plasmid DNAs were verified by DNA sequencing (Eurofins, Huntsville, AL, USA) and transiently transfected into HEK293 cells by electroporation (X-Cell, Bio-Rad, Hercules, CA, USA) as described previously [60].

## Figures and Tables

**Figure 1 ijms-24-03604-f001:**
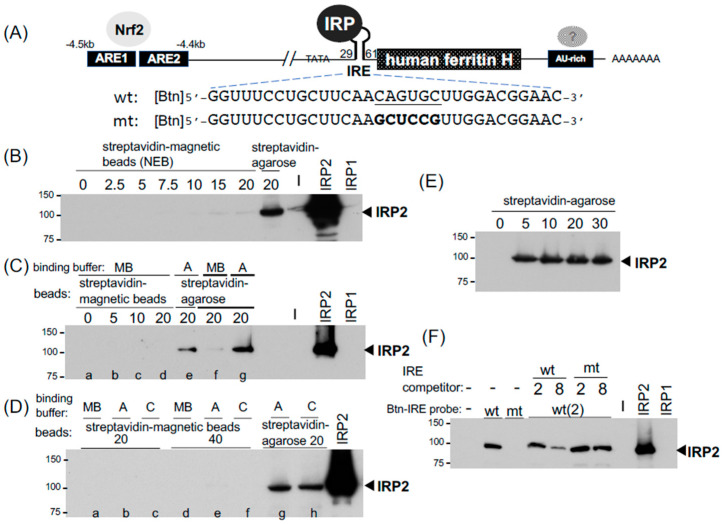
The IRP-IRE pull-down assays and comparison between streptavidin-agarose and -magnetic beads. (**A**) Expression of the human ferritin H gene is regulated at least three cis-acting elements. The wild type (wt) and mutant (mt) IRE (iron responsive element) RNA sequences used for pull-down probes are shown. They were 5’-end biotinylated [Btn]. (**B**) 500 μg of SW480 WCLs (60 μL in RIPA buffer) and 2 μg (equal to ~200 pmol) of biotinylated wt IRE RNA oligonucleotide were incubated in 140 μL of binding buffer A at room temperature for 1 h with constant rotation. 0–20 μL of streptavidin magnetic beads (NEB) or 20 μL of high capacity streptavidin agarose (ThermoFisher Scientific) were washed once with washing buffer, resuspended in 50 μL of binding buffer A, added to the IRE probe/WCLs mixture, and further incubated for 1 hr. The magnetic beads were collected using a magnetic stand, and agarose resins were precipitated by micro-centrifugation at 5000 rpm for 0.5 min. They were washed twice with 1 mL of washing buffer. 12 μL of 2xSDS-PAGE sample buffer was added to precipitated resins, vortexed briefly, and heated at 95 °C for 10 min. After brief spinning, the samples were loaded on 10% SDS-PAGE gel and subjected to IRP2 western blotting. WCLs of HEK293 cells transfected with empty vector, IRP2, or IRP1 expression plasmid was loaded to verify the specificity of anti-IRP2 antibody (**B**–**D**,**F**). (**C**) lanes a–e: 500 μg of SW480 WCLs (50 μL in IP lysis buffer) and 2 μg of biotinylated wt IRE RNA oligonucleotide were incubated in 150 μL of magnetic beads binding buffer (MB, lanes a–d) or binding buffer A (lane e), followed by incubation and pull-down with 0–20 uL of pre-washed streptavidin magnetic beads (lanes a–d) or 20 μL of high capacity streptavidin agarose (lane e). lanes f, g: 500 μg of SW480 WCLs (60 μL in RIPA buffer) and 2 μg of biotinylated wt IRE RNA oligonucleotide were incubated in 140 μL of magnetic beads binding buffer (MB, lane f) or binding buffer A (lane g), pull-down with 20 μL of high capacity streptavidin agarose (lane g), and IRP2 western blotting. (**D**) 500 μg of SW480 WCLs (50 μL in IP lysis buffer) and 2 μg of biotinylated wt IRE RNA oligonucleotide were incubated in 150 μL of magnetic beads binding buffer (MB, lanes a and d), binding buffer A (lanes b, e, and g), or binding buffer C (lanes c, f, and h) followed by pull-down with 20 μL or 40 μL of pre-washed streptavidin magnetic beads (lanes a–f) or 20 μL of high capacity streptavidin agarose (lane g and h), and IRP2 western blotting. (**E**) 250 μg of K562 WCLs (20 μL in IP lysis buffer) and 2 μg of biotinylated wt IRE RNA oligonucleotide were incubated in 180 μL of binding buffer A, and pull-down with 0–30 uL of pre-washed high capacity streptavidin agarose and IRP2 western blotting. (**F**) 500 μg of SW480 WCLs (130 μL in RIPA buffer) and 2 μg of biotinylated wt or mt IRE RNA oligonucleotide were incubated in 370 μL of binding buffer A (total 500 μL) in the absence of presence of 2 μg and 8 μg of non-biotinylated wt and mt RNA oligonucleotide competitors. The binding complex was pulled down with 20 μL of pre-washed high capacity streptavidin agarose and IRP2 western blotting. All experiments were repeated 2–3 times and the representative western blots are shown.

**Figure 2 ijms-24-03604-f002:**
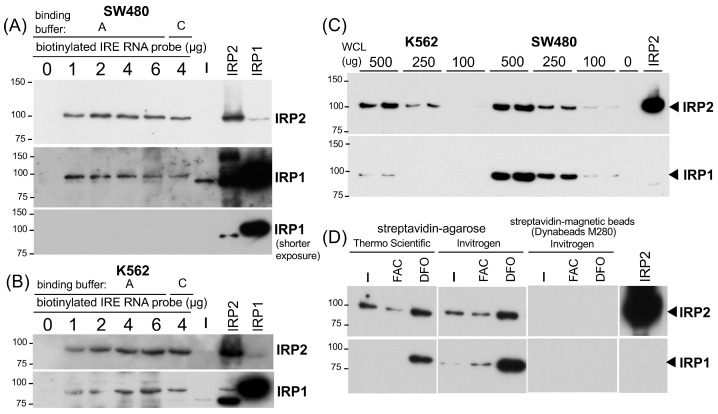
Verification of the IRP-IRE pull-down assay for semi-quantitative detection of IRP2 and IRP1 binding activity. (**A**) 500 μg of SW480 WCLs (30 μL in IP lysis buffer) or (**B**) K562 WCL (40 μL in IP lysis buffer), together with 0–6 μg of biotinylated wt IRE RNA oligonucleotide were incubated in 500 μL of binding buffer A or C. The binding complex was pulled down with 20 μL of pre-washed high capacity streptavidin agarose and subjected to IRP2 and IRP1 western blotting. ECL Clarity and Clarity Max (BIO-RAD) was used for IRP2 and IRP1, respectively. (**C**) 0–500 μg of K562 and SW480 WCLs were incubated with 2 μg of the biotinylated wt IRE RNA oligonucleotide in 200 μL of binding buffer A. The binding complex was pulled down with 20 μL of pre-washed high capacity streptavidin agarose and subjected to IRP2 and IRP1 western blotting. (**D**) 250 μg of WCLs (25 μL in IP lysis buffer) from K562 cells treated with 250 μM FAC or 25 μM DFO for 26 hr were incubated in 150 μL buffer C together with 4 μg of the biotinylated wt IRE probe and 20 μL of high capacity streptavidin agarose (ThermoFisher Scientific), streptavidin agarose (Invitrogen), or 30 μL of Dynabeads M-280 (Invitrogen) simultaneously. The procedures of pull-down/beads wash and western blotting for IRP2 and IRP1 are the same as other experiments. All experiments were repeated 2–3 times and the representative western blots are shown.

**Figure 3 ijms-24-03604-f003:**
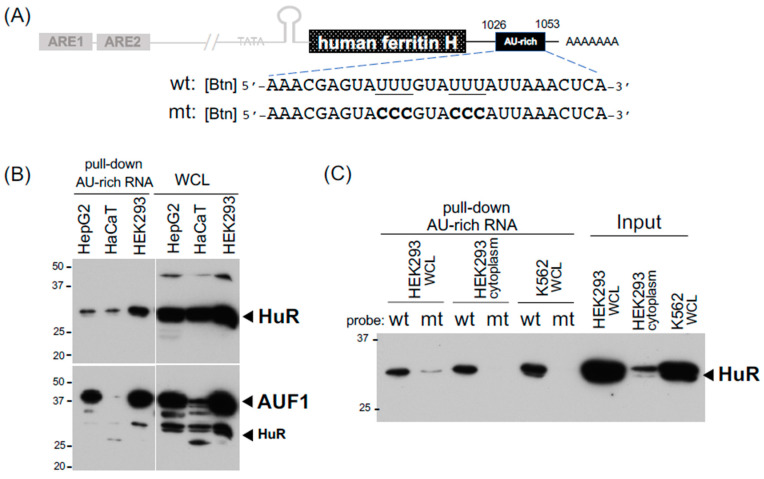
Detection of HuR and AUF1 binding to the 3’-UTR AU-rich element in the human ferritin H mRNA. (**A**) The location and RNA sequences for one of putative AU-rich elements studied in pull-down assays. The RNA probes were 5’-end biotinylated and the mutated AU-rich sequence is also shown. (**B**) 75 μg of WCLs from HepG2, HaCaT, and HEK293 cells were incubated with 4 μg of the 5’-biotinylated wt AU-rich RNA probe and 20 μL of high capacity streptavidin agarose all together for 3 h at room temperature. 20 μg of WCLs were also loaded for input proteins. Western blotting for HuR was done first (12.5% acrylamide gel), followed by incubation with anti-AUF1 antibody. As indicated, some HuR bands came back on the AUF1 western blot. (**C**) 200 μg of WCLs from HEK293 and K562 cells, and 500 μg of the cytoplasmic fraction from HEK293 cells were incubated with 4 μg of the 5’-biotinylated wt or mt AU-rich RNA probe and 20 μL of high capacity streptavidin agarose all together followed by pull-down and HuR western blotting. 30 μg of the WCLs and cytoplasm were loaded for input of HuR. All experiments were repeated 2–3 times and the representative results are shown.

**Figure 4 ijms-24-03604-f004:**
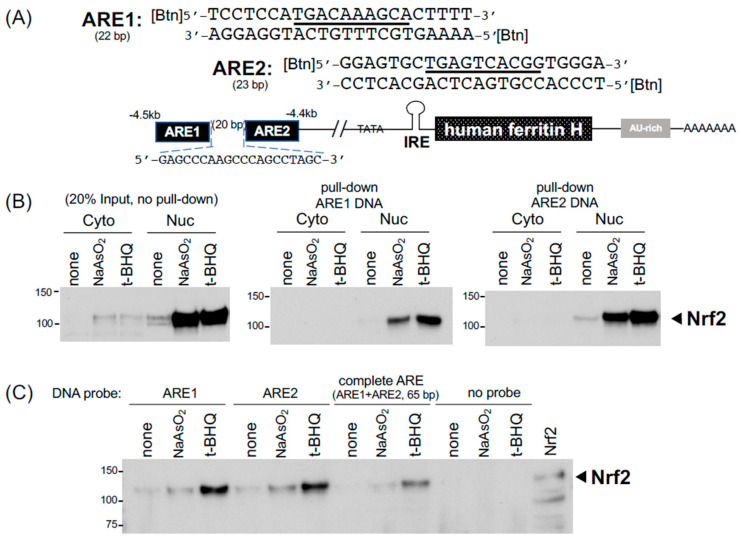
DNA pull-down for detection of Nrf2 binding to the ferritin H antioxidant responsive element (ARE). (**A**) The complete ARE enhancer (65 bp) in the human ferritin H gene is composed of two ARE elements (ARE1 and ARE2) separated by a 20-bp spacer. Their DNA sequences are shown with core binding sites underlined. The double-strand DNA probes were 5’-end biotinylated at both strands. (**B**) K562 cells were treated with 25 μM sodium arsenite or 25 μM t-BHQ for 14 h and the cytoplasmic and nuclear fractions were isolated. 100 μg of the cytoplasmic and nuclear fractions were incubated at room temperature for 2 h with 4 μg of annealed biotinylated ARE1 or ARE2 probe together with streptavidin-agarose in 200 μL of PBS containing 1x inhibitors of proteases (Millipore) and phosphatases (Gold Biotechnology). Pull-down samples were subjected to 10% acrylamide SDS-PAGE and western blotting with anti-Nrf2 antibody (SC-13032). 20 μg of the cytoplasmic and nuclear fractions were also loaded for input of proteins. (**C**) 50 μg of the nuclear fractions isolated from K562 cells treated with 25 μM sodium arsenite or 25 μM t-BHQ for 10 h were incubated at room temperature for 2 h with annealed biotinylated ARE1, ARE2, the complete 65 bp ARE, or no probe together with 20 μL streptavidin-agarose in 200 μL binding buffer C for pull-down the binding complex. Nrf2 western blotting was performed with Nrf2-transfected HEK293 WCL loaded as a positive control. All experiments were repeated 2–3 times and the representative results are shown.

**Table 1 ijms-24-03604-t001:** Chemicals and reagents used in this work.

REAGENT OR RESOURCE	SOURCE	IDENTIFIER (Catalog Number)	LOT Number
** *Primary Antibodies* **			
rabbit monoclonal anti-Aconitase 1 (IRP1)	Abcam	ab126595	YI071316CS
rabbit monoclonal anti-IRP2	Cell Signaling Technology	37135	1
rabbit monoclonal anti-HuRELAVL1	Cell Signaling Technology	12582	1
rabbit monoclonal anti-AUF1/hnRNP D	Cell Signaling Technology	12382	1
rabbit polyclonal anti-Nrf2	Santa Cruz Biotechnology	SC-13032x	I1311
rabbit monoclonal anti-Nrf2	Cell Signaling Technology	12721	8

** *Secondary Antibodies* **			
horse anti-mouse IgG-peroxidase conjugated	Cell Signaling Technology	7076S	31
goat anti-rabbit IgG-peroxidase conjugated	MilliporeSigma	AP132P	

**Streptavidin-resins**			
high capacity streptavidin agarose	ThermoFisher Scientific	20359	QB213813, XB342916
streptavidin agarose	Invitrogen	15942-050	1404248
streptavidin magnetic beads	New England BioLabs	S1420S	10128395
Dyna beads M-280 Streptavidin	Invitrogen	11205D	348667
Dyna beads MyOne Streptavidin T1	Invitrogen	65801D	127408950

** *Culture Media and Supplementals* **			
Dulbecco’s modified Eagle Medium (DMEM)	Corning	50-003-PC	
Minimum Essential Medium (MEM)	Corning	50-011-PC	
RPMI1640	Corning	50-020-PC	
Opti-MEM	Life Technologies	22600-134	
Penicillin Streptomycin solution, 100X	Corning	30-002-CI	
Sodium pyruvate	Corning	MT25000CI	
Non-essential amino acid solution, 100X	Corning	25-025-CI	
Trypsin, 10X	Corning	25-054-CI	
Fetal Bovine Serum	Seradigm	1400-500	

** *Cell Lines* **	** *Source/Culture Media* ** *(containing 1x Penicillin Streptomycin)*		
HaCaT immortalized human keratinocyte	NE Fusenig/ DMEM high glucose (4.5 g/L) +10%FBS		
HepG2 human hepatocellular carcinoma	ATCC/MEM +10% FBS	HB-8065	
HEK293 immortalized human embryonic kidney cells	ATCC/MEM +10% FBS	CRL-1573	
K562 human erythroleukemia	ATCC/RPMI1640, 25 mM HEPES +10%FBS	CCL-243	
SW480 human colon adenocarcinoma	ATCC/DMEM high glucose (4.5 g/L) +10%FBS	CCL-228	

** *Chemicals* **			
Acrylamide	J.T. Baker	4081-01	
Ammonium iron(III) citrate (FAC)	Sigma-Aldrich	F5879	
Ammonium persulfate	Thermo Scientific	17874	
Bis-Acrylamide	EMD Millipore	2620	
Bovine Serum Albumin (BSA) Fraction V	EMD Millipore (Calbiochem)	2930	
Bromophenol blue	Sigma-Aldrich	B-8026	
Clarity Max Western ECL Substrate	Bio-Rad	170-5062	
Clarity Western ECL Substrate	Bio-Rad	170-5061	
Deferoxamine mesylate salt (DFO)	Sigma-Aldrich	D9533	
Glycerol	Fisher Scientific	BP229-1	
Glycine	J.T. Baker	4057-06	
2-mercaptoethanol	EMD Millipore	6050	
Nonidet-P40 (NP40)	US Biological	N-3500	
Pierce ECL Western Blotting substrate	Thermo Scientific	32106	
Phenylmethylsulfonyl fluoride (PMSF)	Sigma-Aldrich	P-7626	
Phosphatase Inhibitors (Simple Stop I)	Gold Biotechnology	GB-450-1	
Potassium Chloride	Fisher Scientific	BP366-500	
Potassium Phosphate, Monobasic	EM Science	B10203-34	
Precision Plus Protein Dual Color Standards	Bio-Rad	161-0374	
Protease Inhibitor Cocktail Set I	Calbiochem/Millipore	539131	
Protein Assay Dye Reagent Concentrate	Bio-Rad	5000006	
Skim milk powder	DIFCO, Beckton-Dickinson	232100	8032502
Sodium Arsenite	Fisher Scientific	S225I-100	51093
Sodium Chloride	BDH	7647-14-5	
Sodium Deoxycholate	Sigma-Aldrich	D-6750	
Sodium Dodecyl Sulfate (SDS)	Calbiochem	7910	
Sodium Phosphate, Dibasic	Fisher Scientific	BP332-500	
TEMED	Fisher Scientific	BP150-100	
tert-Butylhydroquinone (t-BHQ)	Sigma-Aldrich	112941-100G	18830PIHO
Tris (Hydroxymethyl) Aminomethane	J.T. Baker	4109-06	
Tween 20	Fisher Scientific	BP337-500	

** *Binding and Washing Buffer* **			
Binding Buffer A	20 mM Tris pH7.4, 300 mM KCl, 0.2 mM EDTA. 1.5 mM MgCl_2_, and 0.5 mM PMSF		
Binding Buffer C	20 mM Hepes, pH 7.4, 100 mM KCl, 0.5 mM EDTA, 1.5 mM MgCl_2_, 20% glycerol, 1 mM DTT		
Dyna Beads Buffer	5 mM Tris, pH7.5, 1M NaCl, and 0.5 mM EDTA		
Magnetic Beads Buffer (NEB)	20 mM Tris, pH7.5, 0.5M NaCl, and 1 mM EDTA		
Phosphate Buffered Saline (PBS)	137 mM NaCl, 27 mM KCl, 15 mM KH_2_PO_4_, 81 mM Na_2_HPO_4_		
Washing Buffer	25 mM Tris, pH7.4, 15 mM NaCl, 1% NP40, and 0.5% sodium deoxycholate		

** *Lysis Buffer (WCLs)* **			
IP Lysis Buffer	25 mM Tris, pH7.4, 150 mM NaCl, 1 mM EDTA, 1% NP40, and 5% glycerol		
RIPA Buffer	25 mM Tris, pH7.4, 150 mM NaCl, 1% NP40, 0.5% sodium deoxycholate, and 0.1% SDS		

** *Lysis Buffer (Cytoplasm/Nucleus)* **			
Hypotonic Buffer	20 mM Trisl, pH 7.4, 10 mM NaCl, 3 mM MgCl_2_		
Nuclear Extraction Buffer	100 mM Tris, pH 7.4, 2 mM Na_3_VO_4_, 100 mM NaCl, 1% Triton X-100, 1 mM EDTA, 10% glycerol, 1 mM EGTA, 0.1% SDS, 1 mM NaF,		
	0.5% deoxycholate, 20 mM Na_4_P_2_O_7_, and 1X proteinase inhibitor cocktail		

** *Western Blotting Buffer* **			
PVDF Membrane Stripping Solution	1.5% glycine, 0.1% SDS, 1% Tween 20, pH 2.2 adjusted with HCl		
SDS PAGE Running Buffer	25 mM Tris base (no pH adjustment), 192 mM Glycine, and 0.1% SDS,		
2X SDS PAGE Sample Buffer	62.5 mM Tris, pH 6.8, 25% Glycerol, 2% SDS, 0.01% bromophenol blue, and 5% β-mercaptoethanol		
Tris Buffered Saline (TBS)	20 mM Tris, pH 7.6 and 137 mM NaCl		
Western Blotting Transfer Buffer	25 mM Tris pH 8.3, 192 mM Glycine, and 20% methanol		

** *plasmid DNA* **			
pCMVSPORT6 IRP1	Open Biosystems		
pCR4 IRP2	Open Biosystems		
pRK5IRP2	this work		
pRK5Nrf2	this work		

## Data Availability

All data sources presented in this work are directed at the corresponding author Y. Tsuji.

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
