# Peer review of "Optimization of Biotinylated RNA or DNA Pull-Down Assays for Detection of Binding Proteins: Examples of IRP1, IRP2, HuR, AUF1, and Nrf2"

_ijms, 2023, doi:10.3390/ijms24043604_

Round 1

Reviewer 1 Report

Summary: The manuscript by Y. Tsuji is a technical paper that purports to establish the optimal in vitro conditions for evaluating the interaction of nucleic acid binding proteins with specific DNA or RNA sequences via pull-down assay. The general thesis of the work is that a pull-down assay may provide an alternative to a band shift assay in certain experimental situations by allowing for greater flexibility in identifying in vitro conditions amenable to detecting the binding of regulatory proteins to DNA or RNA. The author focuses on the analysis of various proteins that are known or hypothesized to interact with certain DNA or RNA sequences to regulate the expression of the human ferritin H gene at the level of transcription, translation, or mRNA stability. The author assesses different solid-phase streptavidin-coupled matrices (agarose vs. magnetic beads), binding buffers, wash buffers, biotinylated DNA or RNA probes, and cellular extracts to optimize the conditions to specifically detect the interaction of IRP1, IRP2, HuR, AUF1, and Nrf2 with specific RNA or DNA sequences. The author describes presumably the best in vitro conditions to detect the binding of each protein to its respective DNA/RNA target sequence in the ferritin H gene/transcript via pull-down assay.

Critique: The paper describes a hodgepodge of experiments that appeared to be cobbled together for the purpose of showing the utility of pull-down assays to detect the interaction of various regulatory proteins with RNA or DNA sequences in the human ferritin H gene. Hence, there is not much new knowledge generated here and there are a number deficiencies that dampen enthusiasm for the work.

1) The paper suffers from the absence of a more systematic evaluation of how buffer conditions affect the binding of each protein to its respective nucleic acid target. Given that the final detergent and salt concentration conditions differed depending upon the combination of lysis and binding buffers used, it is difficult to infer the relative importance of electrostatic and hydrophobic interactions in stabilizing the various protein-DNA/RNA complexes detected by pull-down assay.

2) The pull-down assay as described herein is inherently qualitative. To achieve a more quantitative assay, the bands on immunoblots ought to be analyzed by densitometry to assess the variability across the three experiments. Even better, a titration study could be done in which the biotinylated DNA or RNA probes are varied over a broader range of concentrations for the purpose of determining EC50s for each target protein. Alternatively, competition assays could be conducted by titrating unlabeled DNA or RNA with a fixed concentration of biotinylated probe to determine IC50s.

3) The concentration of biotinylated RNA or DNA probes used in the pull-down reactions was very high (in the range of 1 uM) whereas the nucleic acid binding affinities of sequence-specific RNA/DNA binding proteins are typically in the nanomolar range. Hence, the assays were likely performed under conditions of excess nucleic acid. While this is understandable if the goal is to merely detect potential nucleic acid binding proteins, it is unlikely that the in vitro conditions reflect the true nature of protein and DNA/RNA target site concentrations as they exist in a cell. These factors are critical to acknowledge in interpreting the effects of mutations in the DNA/RNA probes on the nucleic acid binding properties of each protein.

4) Important controls are missing in some of the figures. For example, input lanes are notably absent in Figs. 1 and 2. Consequently, it is not possible to establish the efficiency of the pull-down reaction with respect to the relative amount of DNA or RNA binding protein present in the cell extract. Mutant probes were not tested in Fig. 4 so it is unclear whether the observed Nrf2-ARE1/2 interactions are specific.

Minor: The apparent inability of streptavidin-coupled magnetic beads to pull-down biotinylated RNA-IRP complexes is puzzling. Paramagnetic beads are often made of magnetite (iron oxide). It is quite possible that leaching of iron ions may explain the unexpectedly negative results seen with the magnetic beads used in the pull-down assays.

Reviewer 2 Report

In this manuscript, the unique author investigates various technical aspects of pull-down experiments to purify RNA and DNA binding proteins using biotinylated DNA and RNA baits. In order to validate his conclusions about this method, he used a set of previously described RNA and DNA binding proteins and their specific targets. Namely, IRP-IRE, ARE-ferritin H, HuR and AUF1 with AU-rich elements.  

The key parameters that were investigated here are, the amount of DNA and RNA baits to use, the lysis and binding buffers, specificity of the detected interactors, the magnetic beads to use and western blot analysis of the characterized proteins. Although the presented results are of general interest for the general RNA biology community, these experiments fall short in one critical issue. This issue is the purity of the pull-downed protein interactors which cannot be assessed by western blot with specific antibodies. Therefore, the resulting pull-downed protein(s) should be analyzed by Mass Spectrometry in order to assess the presence of putative unspecific RNA binders in their sample. This remark applies to both selections using IRE and AU-rich RNAs as baits and the experiment using a DNA molecule as a bait to select Nrf2. This is critical for the specific isolation of previously uncharacterized protein partner.

Specific points:

-       According to the author, why do the magnetic beads fail to efficiently pull-down IRP1 and IRP2? The rationale for this important observation only appears briefly in the Mat and Met section and should be discussed in more details in the main part of the manuscript.   

-       In fig 3B, Jurkat and HL60 WCL are not tested for the presence of HuR and AUF1.  

Typos:

ul should be written m(symbol)L in the whole manuscript.

Reviewer 3 Report

The manuscript titled “Optimization of Biotinylated RNA or DNA Pull-down Assays for Detection of Binding Proteins: Examples of IRP1, IRP2, HuR, AUF1, and Nrf2” by Tsuji, Y. is an original scientific work where the author studies the protein-RNA and protein-DNA interactions by pulling down assays based on biotinylated probes. Five examples were assessed with IRP1, IRP2, AUF1, HuR and Nrf2 proteins to find the most suitable parameters to control the efficiency and quality of DNA/RNA pulling-down assays. The scientific approach and methodology followed by the author seem right and the gathered results can be relevant for the examined field. The knowledge acquired in the present work could significantly aid in the knowledge of the underlying molecular mechanisms that are involved in the binding of RNA/DNA with proteins which can have strong implications in prognosis and the research of human diseases based on genetic disorders. The results achieved are well-discussed during the main body of the reported manuscript. The scientific paper is well written. In my opinion the present manuscript is innovative and the methodological approached used matches with the scope of International Journal of Molecular Sciences. For the above described reasons, I recommend the publication in International Journal of Molecular Sciences once the following remarks will be fixed:

--------

INTRODUCTION

The Introduction is clear and concise. Nevertheless, some minor remarks should be covered in order to improve the scientific quality of the manuscript.

“Since the development of non-radioactive materials for detection of molecular interactions (…) identification of binding proteins by western blotting or preteomic analyses” (lines 42-47). Here, the author should state some recent examples of other techniques reported in bibliography employed to assess biomolecular interactions like bioluminescence energy transfer (BRET) [1], force spectroscopy (FS) [2], molecular recognition imaging (MRI) [3] and fluorescence cross-correlation spectroscopy (FCCS) [4].

[1] Verweij, E.W.E.; et al. BRET-Based Biosensors to Measure Agonist Efficacies in Histamine H1 Receptor-Mediated G Protein Activation, Signaling and Interactions with GRKs and β-Arrestins. Int. J. Mol. Sci. 2022, 23, 3184. https://doi.org/10.3390/ijms23063184.

[2] Pérez-Domínguez, S.; et al. Nanomechanical Study of Enzyme: Coenzyme Complexes: Bipartite Sites in Plastidic Ferredoxin NADP+ Reductase for the Interaction with NADP. Antioxidants 2022, 11, 537. https://doi.org/10.3390/antiox11030537.

[3] Marcuello, C.; et al. Molecular Recognition of Proteins through Quantitative Force Maps at Single Molecule Level. Biomolecules 2022, 12, 594. https://doi.org/10.3390/biom12040594.

[4] Jakobowska, I.; et al. Fluorescence Cross-Correlation Spectroscopy Yields True Affinity and Binding Kinetics of Plasmodium Lactate Inhibitors. Pharmaceuticals 2021, 14, 757. https://doi.org/10.3390/ph14080757.

“This work is focused (…) application of our pull-down assay conditions for identification of proteins interacting with emerging non-coding small RNAs” (lines 95-99). Here, the authors briefly highlights the future potential impact of this work. Nevertheless, it lacks to specify the importance to gain knowledge about the underlying mechanisms hidden of non-coding small RNAs and their interaction with key binding proteins. In this framework, the development of efficient clinical therapies rooted in this technology [5] could work as suitable application.

[5] Smith, E.S.; et al. Clinical Applications of Short Non-Coding RNA-Based Therapies in the Era of Precision Medicine. Cancers 2022, 14, 1588. https://doi.org/10.3390/cancers14061588.

--------

RESULTS

The author perfectly states the most relevant outcomes found in the present work. Some points should be addressed to improve the manuscript quality.

I)        (OPTIONAL) “Binding of (…) recently probed with fluorescent-labeled IRE RNA” (lines 103-105). Here, it may be convenient to introduce some drawbacks related to fluorescent labeled probes like photobleaching effects.

II)     “streptavidin magnetic beads (New England Biolabs (NEB), S140) (…) (ThermoFisher Scientific 20359)” (lines 113-115). This information details should be shifted to the respective Material & Methods sub-section. Moreover, the authors should specify the country of the supplier for each consumable.

III)  “2 µg (…) resuspended in each binding buffer prior to the incubation” (lines 115-121). Similar comment as above. This statement is more appropriate in M&M section because it fully explains the methodology followed by the authors in order to functionalize these magnetic beads.

--------

DISCUSSION

Discussion is well structured. The author perfectly remarks the most relevant insights gathered in this work. Nevertheless, even if it is optional, I may strongly suggest to add a final “Conclusions and future perspectives” section to briefly highlights with few sentences the relevance of this work for all target audiences.

--------

MATERIALS AND METHODS

The manufacturer country should be indicated for each purchased compound/consumable employed by the author.

“acrylamide SDS-PAGE (10% acrylamide, 0.27% bisacrylamide (…))” (lines 429-430). The significant figures should be homogenized. This comment is extendable for the rest of the manuscript body text.

--------

REFERENCES

Bibliography citations are not in the proper format of International Journal of Molecular Sciences. The authors should address this point.

--------

OVERVIEW AND FINAL COMMENTS

The submitted work is well-designed and the gathered results are interesting to optimize those crucial parameters related to DNA/RNA pull-down assays. This fact could positively contribute in this field gaining thus, future impact on society. For this reason, I will recommend the present scientific manuscript for further publication in International Journal of Molecular Sciences once all the aforementioned suggestions will be properly fixed.

Round 2

Reviewer 1 Report

The arguments made in the rebuttal letter and the editorial changes made in the manuscript are acceptable to me.

Reviewer 2 Report

My main concern was the purity of the pull-downed protein interactors. I requested that the purity of these samples has to be analysed by Mass Spectrometry. Unfortunately, although this issue is discussed in the revised manuscript, the authors didn’t perform these experiments.

Round 3

Reviewer 2 Report

This reviewer appreciates the author’s effort for demonstrating the purity of the pulled-downed proteins. As expected, the protein fraction is far from being pure since one can notice many bands on the coomassie-stained membrane although there are much more proteins with the Wt IRE. This confirms that a Mass Spectrometry analysis would add critical informations to this manuscript to carefully assess the purity and the enrichment of the studied proteins. Concerning the time frame, the author could perform an additional pull-down experiment to present this coomassie-stained membrane. Therefore, the author could also send these samples to a Mass Spectrometry facility for analysis of its total protein contents.    
